# Electric Vehicles Selection Based on Brčko District Taxi Service Demands, a Multi-Criteria Approach

Anđelka Štilić [1], Adis Puška [2,*], Aleksandar Đurić [3] and Darko Božanić [4]

1  Academy of Applied Studies Belgrade, The College of Tourism, Bulevar Zorana Đinđića 152a, 11070 Belgrade, Serbia
2  Faculty of Agriculture, Bijeljina University, Pavlovića put bb, 76300 Bijeljina, Bosnia and Herzegovina
3  Faculty of Economics Brčko, University of East Sarajevo, Studentska 11, 76100 Brčko, Bosnia and Herzegovina
4  Military Academy, University of Defence in Belgrade, Veljka Lukica Kurjaka 33, 11000 Belgrade, Serbia
*  Correspondence: adispuska@yahoo.com

**Abstract:** Traditional fuel-powered vehicle emissions have long been recognized as a major barrier to a sustainable environment, and their minimization could ensure both economic support for the sustainable societal fundament and pollution prevention. Electrifying light-duty vehicle fleets, such as taxis, could provide a financial return as well as bring significant economic and environmental improvements. This paper offers a ranked selection of electric vehicles that are presently available on the market, as reviewed by taxi service representatives, as well as their own evaluation of the criteria that influence this selection. This paper provides stability and support when making decisions by deploying stepwise weight assessment ratio analysis and a modified standard deviation method for calculating the subjective and objective weights of the criteria, as well as performing sensitivity analysis to determine how a particular criterion affects the multi-attributive border approximation area. A comparison ranking of the alternatives discovered how a change in the weight value of one of the criteria affected the ranking of the electric vehicle alternatives. According to the research, led by the battery capacity criterion and its values, the Volkswagen ID.3 Pro has the best results and is the taxi of choice in the Brčko District of Bosnia and Herzegovina. Furthermore, the research has demonstrated that the development of electric vehicles for taxi service purposes should strive to extend the range of these vehicles while reducing the battery charging time.

**Keywords:** sustainable transport; electric vehicles; EV; taxi service; SWARA; MSDM; MABAC





## 1. Introduction

In the present era, pollution is strongly affecting the ecosystem, and the notion of sustainable development has become almost a form of "fairy-tale phenomena" [1]. Given the rise of the negative consequences of fossil-fuel transportation and other structures of growth, international authorities and organizations have begun to chart a course toward steady but, more importantly, sustainable growth.

The 2030 Agenda for Sustainable Development [2], which was endorsed by all UN Member States in 2015, presents a shared framework for peace and prosperity for individuals and the planet, both now and in the future. The 17 Sustainable Development Goals (SDGs) are at the core of this initiative, directly and indirectly influenced by the performance of the transport sector [2,3], and represent an urgent demand for action by all economies, both advanced and developing, in a genuine partnership [3,4]. The 2030 Agenda defines 17 SDGs, among which zero hunger (SDG2), good health and well-being (SDG3), affordable and clean energy (SDG7), decent work and economic growth (SDG8), industry, innovation and infrastructure (SDG9), sustainable cities and communities (SDG11), responsible consumption and production (SDG12), and climate action (SDG13), with 169 different targets, are the most dependent on sustainable transport. Following this endorsement, under the Paris Agreement, Bosnia and Herzegovina presented its second national climate

pledge to the United Nations Framework Convention on Climate Change (UNFCCC). The new pledge included emission reduction goals, with transportation as one of the key components [5–9].

The principles and strategies of sustainable development, a green economy, and green growth are interconnected with sustainable transportation, as transportation is a substantial contributor to urban air pollution and reducing urban city pollutants is an utmost necessity. It could be argued that all electric vehicles (EVs), such as battery electric vehicles (BEVs), plug-in hybrid electric vehicles (PHEVs), fuel-cell electric vehicles (FCEVs) [10], etc., are making the planet a healthier place to live by providing a pollution-free form of transportation in busy districts [1]. Although, when talking about "a healthier planet", the full life cycle of EVs could be considered less than optimal, EVs look to be vital to achieving urban air quality, particularly in less-developed economies where the fastest developing districts are among the most polluted [11]. Nonetheless, despite sharp drops in battery costs and hefty tax breaks, EV adoption in those economies is poor. That alone creates a substantial new potential market for taxis and retail delivery vehicles, as they have many times the usage of an ordinary, typical private-use vehicle, and electrifying such vehicles might provide a quick financial return as well as significant economic and environmental improvements. Electrifying taxi networks increases aggregate electricity consumption by approximately 2%, with a negligible rise throughout daily peak usage periods [11]. As a result, the much-desired zero-emission vehicle objectives for commercial light-duty vehicle fleets appear to be ecologically, economically, and socially sound.

Accordingly, the objective of this paper is to provide a ranked selection of EVs that are currently available on the market, as reviewed by taxi service representatives, along with their own evaluation of the variables that influence this selection, and also to help achieve stability in decision-making when applying multi-criteria decision-making (MCDM) methods through giving support to decision-makers when purchasing EVs. This paper seeks to provide stability and support when making decisions, given the example of buying an EV for the use of the taxi service in the Brčko District of Bosnia and Herzegovina; the contributions of this paper could be reflected in the application of innovative decision-making modeling.

The structure of the paper consists of five sections, following the introduction. The literature review in Section 2 of this paper presents the previous works published on the subject of EV usage in commercial light-duty vehicle fleets and the MCDM approaches that were used in the selection of the different EVs. The research methodology is described in Section 3, which also includes a presentation of the methodology and an explanation of the research methods, followed by the research results that are presented in Section 4, where the most significant findings are offered. In the discussion that follows in Section 5, the results are further reviewed and compared with findings from related research. The most significant findings, limitations, and suggestions for future research will be highlighted in Section 6—the conclusion of this paper.

## 2. Literature Review

The literature review will first address the core selection of EVs in taxi services, and second, the use of the different MCDM methods employed for their selection, as the focus of this research is on the choosing of EVs using the MCDM approach.

### 2.1. The Use of EVs in Taxi Services

The use of EVs is encouraged in developing countries since traditional fuel vehicle emissions have long been recognized as a major barrier to a sustainable environment. Anand et al. [12] highlight that traditional vehicle minimization could ensure economic support for society, as well as pollution reduction. Subsidies that encourage the adoption of EVs, as discussed by Sheldon and Dua [10], are an important part of the overall goal to reduce air pollution and greenhouse gas emissions from the light-duty commercial vehicle industry.

The research regarding the use of EVs in taxi services is extensive and ranges widely from the transition from petrol to electric engines to the different charging issues. As a starting point, the transition from traditional petrol and diesel-based internal combustion vehicles to EVs was researched by many authors. Tseng et al. [13] investigated the sustainability of employing electric taxis by optimizing the taxi service strategy. They compared it to traditional vehicles powered by internal combustion engines. Lu et al. [14] designed the operation of taxi services, where there is a combination of petrol and electric vehicles in use. Optimization was used in this case to reduce operational expenses through the application of heuristic solutions. Tamis and van den Hoed [15] researched taxi drivers' readiness to adopt EVs. They focused on the differences between those who were interested in buying an EV compared to those who were not. Baek et al. [16] investigated the usage of EVs in Daejeon, South Korea for taxi-service demands. They achieved this by relying on expert decision-making, based on an in-depth interview, and concluded that to use EVs in taxi services, the infrastructure of charging stations for EVs in South Korea must be expanded. Scorrano et al. [17] interviewed taxi license-holders in Florence to assess the implementation of the EV strategy in taxi service operations. On that occasion, they obtained results demonstrating that the usage of EV taxis is a viable option not only from an ecological standpoint but also from an economics perspective.

Numerous authors addressed the different EV charging issues, while La Rocca and Cordeau [18] examined the adoption of EVs by taxi services, using Teo Taxi as an example. They were looking to address three operational issues: uploading, moving and charging the EV taxis. Jung et al. [19] researched the utility EV for taxi service requirements and determined that the most significant limitation to adoption is the charging of these vehicles. As a result, they created a simulation that suggested that using a shared taxi might be a viable option. Lokhandwala and Cai [20] addressed the challenge of optimizing the EV charging infrastructure. This optimization might aid in avoiding traffic congestion, particularly for taxi drivers in New York City. Cilio and Babacan [21] conducted a simulation to assess the number of EV charging base stations required in Istanbul to convert all taxis to EVs. They concluded that 1363 to 1834 charging stations were required in the Istanbul area at the time. Fraile-Ardanuy et al. [22] first analyzed how much energy was required for EVs in taxi services, and then proposed the optimization of charging stations based on those data. Furthermore, they demonstrated that the expense of purchasing an EV is greater than that of a traditional vehicle.

Some of the reviewed articles addressed issues that are still in the future if we consider the local infrastructure. Li et al. [23] investigated the feasibility of charging up electric taxis while driving. The issue with this type of charging is that it makes invoicing difficult; therefore, they attempted to address that problem in their paper. Vaidya and Mouftah [24] investigated the feasibility of deploying self-driving EVs to meet the demands of taxi services. They explored the idea of wireless charging in order to increase the vehicle's range, and they were also working on a solution to the problem of taxi delays. Vaidya and Mouftah [25] argued for the deployment of more autonomous electric vehicles in order to establish the Internet of Vehicles, which will allow EVs to connect with one another in order to gather necessary information and would make it possible to administer taxi-charging services automatically in this manner.

### 2.2. Application of MCDM Approaches in the Selection of Electric Vehicles

As EVs could be characterized by many parameters, such as horsepower, maximum torque, battery capacity, charging time, range, price [26,27], etc., evaluating and selecting EVs comprise a multi-criteria problem that is characterized by uncertainty and requires looking for Pareto-optimal solutions [27]. Our review of the research papers that address the topic of the application of MCDM methods in the selection of EVs is presented in Table 1. The literature review focuses on the research published in the past three years, with an emphasis on the method(s) used.

**Table 1.** The application of multi-criteria decision-making (MCDM) methods in the selection of EVs.

| Author(s) | Year | Article Title | Method(s) |
|---|---|---|---|
| Biswas and Das [28] | 2019 | Selection of Commercially Available Electric Vehicle using Fuzzy AHP-MABAC | AHP MABAC |
| Biswas et al. [29] | 2019 | An Integrated Methodology for the Evaluation of Electric Vehicles under a Sustainable Automotive Environment | CoCoSo CRITIC |
| Khan et al. [30] | 2020 | Sustainable Hybrid Electric Vehicle Selection in the Context of a Developing Country | TOPSIS |
| Biswas et al. [31] | 2020 | Selection of Commercially Available Alternative Passenger Vehicles in the Automotive Environment | CoCoSo CRITIC |
| Büyüközkan and Uztürk [32] | 2020 | Fleet Vehicle Selection for Sustainable Urban Logistics | SAW VIKOR |
| Ziemba [27] | 2020 | Multi-Criteria Stochastic Selection of Electric Vehicles for the Sustainable Development of Local Government and State Administration Units in Poland | PROSA Monte Carlo |
| Ali et al. [33] | 2020 | Development of a New Hybrid Multi-criteria Decision-making Method for a Car Selection Scenario | TOPSIS FCF-TOPSIS AHP |
| Sonar and Kulkarni [34] | 2021 | An Integrated AHP-MABAC Approach for Electric Vehicle Selection | AHP MABAC |
| Oztaysi et al. [35] | 2021 | Electric Vehicle Selection by Using Fuzzy KEMIRA | KEMIRA |
| Cakir et al. [36] | 2021 | Neutrosophic Fuzzy MARCOS Approach for Sustainable Hybrid Electric Vehicle Assessment | MARCOS |
| Ziemba [37] | 2021 | Selection of Electric Vehicles for the Needs of Sustainable Transport under Conditions of Uncertainty—A Comparative Study on Fuzzy MCDA Methods | TOPSIS SAW NEAT F-PROMETHEE II |
| Ziemba [38] | 2021 | Multi-Criteria Approach to Stochastic and Fuzzy Uncertainty in the Selection of Electric Vehicles with High Social Acceptance | NEAT F-PROMETHEE Monte Carlo SMAA |
| TEPE [39] | 2021 | The Interval-Valued Spherical Fuzzy-Based Methodology and Its Application to Electric Car Selection | IVSF AHP ELECTRE |
| Oztaysi [40] | 2022 | Electric Vehicle Selection by Using Fuzzy SMART | SMART |
| Wei and Zhou [41] | 2022 | Multi-Criteria Decision-Making Framework for the Electric Vehicle Supplier Selection of Government Agencies and Public Bodies in China | BWM VIKTOR |
| Stopka et al. [42] | 2022 | Application of Multi-Criteria Decision-Making Methods for Evaluation of Selected Passenger Electric Cars: A Case Study | Basic variant method AHP |

AHP—Analytic Hierarchy Process, MABAC—Multi Attributive Border Approximation Area Comparison, CoCoSo—Combined Compromise Solution, CRITIC—Criteria Importance Through Inter-criteria Correlation, TOPSIS—Technique for Order Preference by Similarity to the Ideal Solution, SAW—Simple Additive Weighting, VIKTOR—VIseKriterijumska Optimizacija I Kompromisno Resenje, PROSA—Product Sustainability Assessment, FCF-TOPSIS—Full Consistency Fuzzy Technique for Order Preference by Similarity to the Ideal Solution, KEMIRA—Kemeny Median Indicator Ranks Accordance, MARCOS—Measurement of Alternatives and Ranking according to Compromise Solution, NEAT F-PROMETHEE—New Easy Approach To Fuzzy Preference Ranking Organization Method for Enrichment Evaluation, SMAA—Stochastic Multicriteria Acceptability Analysis, IVSF—Interval-Valued Spherical Fuzzy Sets, ELECTRE—Elimination et Choix Traduisant La Réalité, SMART—Simple Multi-Attribute Rating Technique, BWM—Best-Worst Method.

In different conditions, the reviewed literature [27–42] confirmed that criteria such as vehicle pricing, accelerating time, battery range and charge time, top speed, and so forth, are utilized regardless of the MCDM method(s). On the other hand, the fact that different approaches may provide contradictory findings, such as the different recommended models

of EVs, is clearly a weakness in the referenced research; therefore, additional academic discussion is welcomed.

### 3. Methodology

In order to perform this research, it is initially important to comprehend the MCDM implementation postulates. When numerous alternatives are evaluated using certain criteria, MCDM is to be performed and, in order to evaluate the alternatives, it is required that we first identify the criteria. In this section, the methods used to obtain the data and to process it using MCDM approaches will be described.

This research will consist of the following phases [43,44]:

- Phase 1. Determination of alternatives and criteria.
- Phase 2. Data collection and the taxi drivers' evaluation of the criteria.
- Phase 3. Calculation of the subjective weights of the criteria.
- Phase 4. Formation of the primary decision matrix.
- Phase 5. Calculation of the objective weight and the final weight of the criteria.
- Phase 6. Ranking of alternatives.
- Phase 7. Conducting a sensitivity analysis.

Phase 1 of this research is to identify the alternatives and criteria for selecting an EV for the purposes of taxi services in the Brčko District. Due to the numerous vehicle manufacturers and available models on the market, it was necessary to first select the alternatives that would be included in this research. First, taxi drivers who would assist in this research have been identified. Twelve taxi drivers were interviewed, and the alternatives and criteria were determined in collaboration with them. The interview was structured in such a way that the alternatives were initially discussed with taxi drivers. First, they were queried about the constraints that, in their judgment, the alternatives should satisfy. Their responses were systematized, and four constraints were identified, based on which, 11 of the initial 30 alternatives were chosen. A semi-structured interview was conducted on that occasion. The taxi drivers were then asked, in a similar manner, which would be the most crucial requirements for them to purchase an EV. Based on their responses, ten criteria were selected as the most essential from the taxi driver's perspective, and these criteria will be employed in this research. The limitations for alternative selection were established as well. These limitations are as follows:

- The EVs under consideration must have authorized service centers in the area of Brčko District;
- EVs must originate from manufacturers with which taxi drivers are already familiar and have prior expertise;
- EVs must transport at least 5 passengers;
- EVs must be priced at a maximum of EUR 40,000, considering that Bosnia and Herzegovina is a developing nation with limited financial resources, including those for taxi drivers.

Based on these limitations, the following EV alternatives were taken into account in this research, namely: Opel Corsa-e (A1), Nissan Leaf (A2), Peugeot e-208 (A3), Mazda MX-30 (A4), Hyundai Kona Electric (A5), Renault Mégane E-Tech EV40 (A6), Citroen e-C4 (A7), Kia e-Soul (A8), Opel Mokka-e (A9), Peugeot e-2008 (A10), and Volkswagen ID.3 Pro (A11).

Following alternative identification, the criteria were established in collaboration with the same taxi drivers (Table 2).

For the evaluation of alternatives, 10 criteria were selected. Since the criteria differ, it is required that we identify the types of criteria. If there is a specific criterion for taxi drivers that should have higher values, that criterion is a benefit-type criterion, e.g., the vehicle's total range on a single charge should be as high as possible. If there is a specific criterion for taxi drivers that should have lower values, the criterion is a cost-type criterion, e.g., the charging speed of the battery should be as slow as possible.

**Table 2.** The criteria used to evaluate EVs.

| ID | Criterion | Abvr. | Description | Unit | Reference | Criterion Type |
|---|---|---|---|---|---|---|
| C1 | Acceleration 0–100 km/h | ACC | Acceleration from 0 to 100 km/h | s | Hinov et al. [45], Ecer [46] | cost |
| C2 | Top Speed | TS | Maximum EV speed | km/h | Naumovich et al. [47], Du et al. [48] | benefit |
| C3 | Total Power | TP | Total engine power | hp | Bessler et al. [49], Ziemba [27] | benefit |
| C4 | Total Torque | TT | Engine torque | Nm | Ecer [44], Li [50] | benefit |
| C5 | Battery Capacity | BC | Battery capacity | KW | Zhu et al. [51], Ziemba [27] | benefit |
| C6 | Charge Time | CT | Battery charging time in minutes using a standard outlet | min | Lucas et al. [52], Sonar and Kulkarni [34] | cost |
| C7 | Fast-charge Time | FT | Battery charging time in minutes using a fast charger | min | Figenbaum [53], Ecer [46] | cost |
| C8 | Range | R | Full battery range | km | Yang et al. [54], Sonar and Kulkarni [34] | benefit |
| C9 | Price | P | EV value expressed in Euro currency | € | Noel et al. [55], Ecer [46] | cost |
| C10 | Cargo Volume | CV | Total trunk volume | L | Ziemba [27], Eslaminia and Azimi [56] | benefit |

Phase 2. After establishing the alternatives and criteria, it was necessary to collect data on all alternatives for the criteria set [57]. The ev-database.org site (accessed on 19 September 2022), which features the majority of EVs, along with their characteristics, was used to collect data on alternatives and feature values. Following that process, data were collected from taxi drivers. A survey questionnaire for the taxi drivers was designed in order to evaluate the importance of a specific criterion for purchasing an EV. Alternatives and criteria for this research were developed based on the prior interview and the taxi drivers were given a questionnaire in which they rated these criteria. Taxi drivers were asked to rank the criteria on a scale of 1 to 5 with 1 indicating that the criterion was unimportant to them and 5 indicating that it was extremely significant when purchasing an EV. This research was conducted with taxi drivers in person and they solely evaluated the criteria at this point.

Phase 3 is used to assign subjective weights to the criteria. The weights of the criteria will be derived based on the ranking and the SWARA (stepwise weight assessment ratio analysis) method, introduced by Keršuliene et al. [58]. The five steps of the SWARA method will serve to describe this phase.

Step 1. Calculating the mean values of criterion weights, based on the taxi drivers' rankings.

Step 2. Classification of the criteria according to their significance, from the highest to the lowest mean values.

Step 3. Determining the relative importance of criteria. The criterion with the greatest importance is assigned a value of one (1), and the relative importance of the criteria is determined using the mean value of the criteria. The coefficient $k_j$ is determined as the difference between the criteria, added to the number 1. This is accomplished using the following expression:

$$k_j = \begin{cases} 1 \ if \ j = 1 \\ s_j + 1 \ if \ j > 1 \end{cases}. \tag{1}$$

Step 4. Recalculating the relative importance as $q_j$. To calculate the value of these criteria, the $k_j$ value is divided by the $q_j$ value of the previous criterion. It is necessary to

state that $k_j = q_j = 1$ is always the value of the first criterion. This is accomplished with the following expression:

$$q_j = \begin{cases} 1 \ if \ j = 1 \\ \frac{q_j - 1}{k_j} \ if \ j > 1 \end{cases}.$$ (2)

Step 5. Calculating the criteria's weight. Here, individual $q_j$ values are divided by the sum of the $q_j$ values for all criteria. This is accomplished using the following expression:

$$w_j = \frac{q_j}{\sum_{j=1}^{n} q_k}.$$ (3)

Phase 4. The subjective weights of the criterion are calculated by using the SWARA method steps and taxi driver feedback. To calculate the objective weights of the criteria and perform the EV ranking, a primary decision-making matrix must first be created. The primary decision matrix is a tabular presentation of the value of alternatives, based on specific criteria. This matrix is formed using data collected on the vehicle characteristics. This decision matrix is then normalized. Normalization ensures that all criteria are comparable and prepared for further processing [59].

Phase 5. To calculate the objective weights, one of the methods for determining these weights must be employed. The MSDM (modified standard deviation method), developed by Puška et al. [60], will be used to calculate the objective weights in this research. The MSDM method achieves results that are similar to other methods used for objectively determining the weights of the criteria; however, it is simpler to execute, and the steps are as follows:

Step 1. Performing the normalization of the primary decision matrix. Normalization is calculated based on the following expressions:

$$n_{ij} = \frac{x_{ij}}{x_{j \ max}}, \text{ for the benefit criteria}$$ (4)

$$n_{ij} = \frac{x_{j \ min}}{x_{ij}}, \text{ for the cost criteria.}$$ (5)

Step 2. Calculation of the standard deviation values for criteria.

Step 3. Calculation of the sum of the normalized criteria values, $\sum_{j}^{m} x_{ij}$.

Step 4. Calculation of the modified standard deviation value. This is performed so that the standard deviation value can be divided by the sum of the normalized values of the criteria. This step is performed using the following expression:

$$\sigma' = \frac{\sigma}{\sum_{j}^{m} x_{ij}}.$$ (6)

Step 5. Calculation of criteria weights:

$$w_j = \frac{\sigma'_j}{\sum_{j=1}^{m} \sigma'_j}.$$ (7)

By applying the formulas of the MSDM method, the objective weights of the criteria are determined. The final weights of the criteria are formed in such a way that the average weight for a specific criterion is determined, based on the subjective and objective weights of the criteria. Since the sum of the weights' values can be lower or greater than 1, the individual weights of the criteria must be divided by the sum of the weights. As a result, the aggregate weight values will equal 1. The subjective and objective weights are used since this takes into consideration the respondents' opinions; by using more objective weights, the weights are determined based on the difference in the values of those criteria.

If there are larger deviations in the values of one criterion, the weight of that criterion will be higher, and vice versa. In this manner, preference is given to those criteria with a greater deviation and those where the values differ from those of other criteria. The advantages of determining weights by subjective and objective methods are expressed by employing the final weight of the criteria.

Phase 6. The EV alternatives are ranked after the primary decision-making matrix has been formed and the weights of the criteria have been set. The purpose of the ranking of the alternatives is to provide a ranking order of alternatives based on a compromise ranking. When one of the alternatives does not offer all the best characteristics, a compromise is made when calculating the ranking order. To be highly ranked, a particular alternative must meet as many of the specific criteria as possible. To rank the alternatives, generate a ranking order, and identify which of the alternatives best meets the decision criteria set, the MABAC (multi-attributive border approximation area comparison) method, developed by Pamučar and Ćirović [61], will be used. This method follows the calculation steps as listed:

Step 1. Formation of the initial decision matrix.

Step 2. Normalization of the initial decision matrix. The elements of the initial decision matrix are normalized, according to the following expressions:

$$n_{ij} = \frac{x_{ij} - x_i^-}{x_i^+ - x_i^-} \text{ for the benefit criteria} \tag{8}$$

$$n_{ij} = \frac{x_{ij} - x_i^+}{x_i^- - x_i^+} \text{ for the cost criteria.} \tag{9}$$

Step 3. Calculation of indicators of the weighted matrix ($V$) elements. This indicator is calculated using the following expression:

$$\widetilde{V}_{ij} = w_i \cdot \widetilde{n}_{ij} + w_i \tag{10}$$

This method is distinguished by the inclusion of an additional criteria weight in the weighted matrix.

Step 4. Determination of the border approximate area matrix ($G$). The formula for the geometric mean is used to calculate this value. This indicator is calculated using the following expression:

$$G = \left( \prod_{j=1}^{m} \widetilde{v}_{ij} \right)^{1/m} \tag{11}$$

Step 5. Calculating the distance of the elements of the weighted matrix ($V$) from the value of the approximate border area matrix ($G$). This step is calculated using the following expression:

$$\widetilde{Q} = \widetilde{V} - \widetilde{G}. \tag{12}$$

Step 6. Ranking alternatives. The alternatives are ranked according to the MABAC method's value, with the best alternative having the highest MABAC method value. The MABAC method is unique in the sense that its values might even be negative, i.e., for alternatives with values lower than the approximate border area matrix ($G$).

Phase 7. A sensitivity analysis will be conducted to determine how a particular criterion affects the ranking of alternatives. The purpose of this analysis is to discover how a change in the weight value of one of the criteria affects the ranking of the alternatives. This will indicate how "sensitive" any of the alternatives are to a certain criterion.

In order to conduct the sensitivity analysis, the method of gradually reducing the weight of each individual criterion will be utilized [62–68]. Each of the criteria's weights will be reduced by 15%. For the C1 (acceleration) criterion, for instance, the weight will initially drop by 15%, then by another 15%, until the criterion's value is reduced to 10%. This amount can only be obtained by reducing a certain criterion by six times. The sensitivity analysis will be conducted using 60 scenarios, which will be created as a result of the 10 criteria that

were used in this research. By using this type of sensitivity analysis, it is possible to identify which criterion—and to what extent—is responsible for the ranking order.

## 4. Results

Calculating the research results involves determining the criteria weights, in order to rank the alternatives. First, the subjective criteria weights will be determined. The preselected taxi drivers rated how significant the criteria were to them when selecting an EV for utilization in the taxi service (Table 3). They assigned grades ranging from 1 to 5. Grade 1 denotes the least significance, while grade 5 denotes the most significance.

**Table 3.** Taxi drivers' grading.

|  | C1 (ACC) | C2 (TS) | C3 (TP) | C4 (TT) | C5 (BC) | C6 (CT) | C7 (FT) | C8 (R) | C9 (P) | C10 (CV) |
|---|---|---|---|---|---|---|---|---|---|---|
| T1 | 3 | 3 | 5 | 5 | 5 | 5 | 5 | 5 | 5 | 3 |
| T2 | 3 | 3 | 4 | 3 | 4 | 4 | 4 | 5 | 4 | 4 |
| T3 | 2 | 1 | 4 | 4 | 5 | 5 | 5 | 5 | 5 | 3 |
| T4 | 5 | 3 | 4 | 5 | 5 | 5 | 5 | 5 | 3 | 4 |
| T5 | 2 | 1 | 3 | 3 | 5 | 5 | 5 | 5 | 5 | 4 |
| T6 | 5 | 3 | 3 | 5 | 3 | 3 | 4 | 3 | 2 | 2 |
| T7 | 3 | 4 | 5 | 2 | 5 | 4 | 5 | 5 | 5 | 2 |
| T8 | 1 | 2 | 3 | 2 | 5 | 5 | 5 | 5 | 5 | 4 |
| T9 | 1 | 1 | 4 | 3 | 5 | 5 | 5 | 4 | 5 | 3 |
| T10 | 3 | 4 | 4 | 3 | 5 | 5 | 5 | 5 | 5 | 3 |
| T11 | 1 | 1 | 4 | 3 | 5 | 5 | 5 | 4 | 5 | 5 |
| T12 | 2 | 2 | 3 | 1 | 4 | 5 | 4 | 5 | 5 | 3 |
| Average | 2.58 | 2.33 | 3.83 | 3.25 | 4.67 | 4.67 | 4.75 | 4.67 | 4.50 | 3.33 |
| Range | 9 | 10 | 6 | 8 | 2 | 2 | 1 | 2 | 5 | 7 |

Following the collection of taxi driver ratings, the average ratings were determined, and the criteria were ranked based on that rating. Considering that these are the first two steps in bringing the SWARA method into practice, these calculations must be completed. Following these criteria rankings, the weights of the criteria are calculated (Table 4). Based on these findings, it is clearly crucial for taxi drivers to be able to charge their EV batteries as rapidly as possible so that they can keep working. Fast charge time, criterion C7, was assigned the most weight and significance. The next group of criteria was C5 (battery capacity), C6 (charge time), and C8 (range), all of which were assigned the same weight ($w = 0.1549$). The lowest ranking was assigned to criterion C2 (top speed). This is due to the fact that the majority of taxi rides are provided in areas with speed restrictions, and the "maximum speed" criterion does not make a point of argument when speed cannot be achieved, due to such constraints.

**Table 4.** SWARA-based criteria weight calculation.

| Criteria | $s_j$ | $k_j$ | $q_j$ | $w_j$ |
|---|---|---|---|---|
| C7 (FT) |  | 1.0000 | 1.0000 | 0.1678 |
| C5 (BC) | 0.0833 | 1.0833 | 0.9231 | 0.1549 |
| C6 (CT) | 0.0000 | 1.0000 | 0.9231 | 0.1549 |
| C8 (R) | 0.0000 | 1.0000 | 0.9231 | 0.1549 |
| C9 (P) | 0.1667 | 1.1667 | 0.7912 | 0.1328 |
| C3 (TP) | 0.6667 | 1.6667 | 0.4747 | 0.0797 |
| C10 (CV) | 0.5000 | 1.5000 | 0.3165 | 0.0531 |
| C4 (TT) | 0.0833 | 1.0833 | 0.2921 | 0.0490 |
| C1 (ACC) | 0.6667 | 1.6667 | 0.1753 | 0.0294 |
| C2 (TS) | 0.2500 | 1.2500 | 0.1402 | 0.0235 |
|  |  | sum | 5.9594 |  |

To determine the objective weights of the criteria using the MSDM approach, an initial decision matrix must be created once the subjective weights have been determined. This initial decision matrix is also necessary to rank the alternatives. The criteria values for each alternative are inserted to shape the initial decision-making matrix. As can be observed, this decision matrix's values (Table 5) are much more wide-ranging since various measurement units are employed. In order to make the data eligible for further analysis, the data must be normalized.

**Table 5.** Initial decision matrix.

| | C1 (ACC) | C2 (TS) | C3 (TP) | C4 (TT) | C5 (BC) | C6 (CT) | C7 (FT) | C8 (R) | C9 (P) | C10 (CV) |
|---|---|---|---|---|---|---|---|---|---|---|
| A1 | 8.1 | 150 | 100 | 260 | 50.0 | 435 | 26 | 285 | 32,895 | 309 |
| A2 | 7.9 | 144 | 110 | 320 | 40.0 | 765 | 43 | 235 | 32,940 | 435 |
| A3 | 8.1 | 150 | 100 | 260 | 50.0 | 435 | 26 | 285 | 31,050 | 265 |
| A4 | 9.7 | 140 | 107 | 271 | 35.5 | 195 | 39 | 170 | 34,990 | 366 |
| A5 | 9.9 | 155 | 100 | 395 | 42.0 | 390 | 47 | 250 | 35,495 | 332 |
| A6 | 10.0 | 150 | 96 | 250 | 40.0 | 135 | 29 | 250 | 35,200 | 440 |
| A7 | 9.7 | 150 | 100 | 260 | 50.0 | 435 | 26 | 265 | 36,140 | 380 |
| A8 | 9.9 | 157 | 100 | 395 | 42.0 | 390 | 47 | 230 | 33,495 | 315 |
| A9 | 9.2 | 150 | 100 | 260 | 50.0 | 435 | 26 | 255 | 37,650 | 310 |
| A10 | 8.5 | 150 | 100 | 260 | 50.0 | 435 | 26 | 255 | 36,330 | 434 |
| A11 | 7.3 | 160 | 150 | 310 | 62.0 | 375 | 33 | 350 | 38,060 | 385 |

After the initial decision-making matrix is formed, the weights of the criteria are calculated according to the MSDM method. Data normalization is the first step of this method. Thereafter, the standard deviation (SD) value, as well as the sums of the values of the columns, that is, the criteria (sum), are calculated. The next step is the formation of a modified SD, which is obtained by dividing the values of the SD by the sums of the columns. After that, step 5 is performed, and the weights of the criteria are calculated. The results reveal (see Table 6) that criteria C6 (charge time) had the highest value since its values were the most varied across the alternatives. Criteria C7 and C4 then follow it, with criterion C2 (top speed) having the lowest weight value due to its lowest dispersion.

**Table 6.** The MSDM method's weighting results.

| | C1 (ACC) | C2 (TS) | C3 (TP) | C4 (TT) | C5 (BC) | C6 (CT) | C7 (FT) | C8 (R) | C9 (P) | C10 (CV) |
|---|---|---|---|---|---|---|---|---|---|---|
| A1 | 0.9012 | 0.9375 | 0.6667 | 0.6582 | 0.8065 | 0.3103 | 1.0000 | 0.8143 | 0.9439 | 0.7023 |
| A2 | 0.9241 | 0.9000 | 0.7333 | 0.8101 | 0.6452 | 0.1765 | 0.6047 | 0.6714 | 0.9426 | 0.9886 |
| A3 | 0.9012 | 0.9375 | 0.6667 | 0.6582 | 0.8065 | 0.3103 | 1.0000 | 0.8143 | 1.0000 | 0.6023 |
| A4 | 0.7526 | 0.8750 | 0.7133 | 0.6861 | 0.5726 | 0.6923 | 0.6667 | 0.4857 | 0.8874 | 0.8318 |
| A5 | 0.7374 | 0.9688 | 0.6667 | 1.0000 | 0.6774 | 0.3462 | 0.5532 | 0.7143 | 0.8748 | 0.7545 |
| A6 | 0.7300 | 0.9375 | 0.6400 | 0.6329 | 0.6452 | 1.0000 | 0.8966 | 0.7143 | 0.8821 | 1.0000 |
| A7 | 0.7526 | 0.9375 | 0.6667 | 0.6582 | 0.8065 | 0.3103 | 1.0000 | 0.7571 | 0.8592 | 0.8636 |
| A8 | 0.7374 | 0.9813 | 0.6667 | 1.0000 | 0.6774 | 0.3462 | 0.5532 | 0.6571 | 0.9270 | 0.7159 |
| A9 | 0.7935 | 0.9375 | 0.6667 | 0.6582 | 0.8065 | 0.3103 | 1.0000 | 0.7286 | 0.8247 | 0.7045 |
| A10 | 0.8588 | 0.9375 | 0.6667 | 0.6582 | 0.8065 | 0.3103 | 1.0000 | 0.7286 | 0.8547 | 0.9864 |
| A11 | 1.0000 | 1.0000 | 1.0000 | 0.7848 | 1.0000 | 0.3600 | 0.7879 | 1.0000 | 0.8158 | 0.8750 |
| SD | 0.0944 | 0.0346 | 0.1012 | 0.1377 | 0.1193 | 0.2329 | 0.1952 | 0.1251 | 0.0560 | 0.1354 |
| sum | 9.0887 | 10.350 | 7.7533 | 8.2051 | 8.2500 | 4.4728 | 9.0621 | 8.0857 | 9.8122 | 9.0250 |
| MSD | 0.0104 | 0.0033 | 0.0131 | 0.0168 | 0.0145 | 0.0521 | 0.0215 | 0.0155 | 0.0057 | 0.0150 |
| $w_o$ | 0.0619 | 0.0199 | 0.0778 | 0.1000 | 0.0862 | 0.3103 | 0.1283 | 0.0922 | 0.0340 | 0.0894 |

The average weights of the criteria are calculated once the subjective and objective weights have been obtained. It was not necessary to correct the criteria weights since

the sum of the obtained weights was equal to 1. The results (Table 7) reveal that C6 was given the highest weight, followed by C7, C8, and C5. The lowest weight was given to criterion C2.

**Table 7.** The final weights of the criteria.

|  | C1 (ACC) | C2 (TS) | C3 (TP) | C4 (TT) | C5 (BC) | C6 (CT) | C7 (FT) | C8 (R) | C9 (P) | C10 (CV) |
|---|---|---|---|---|---|---|---|---|---|---|
| $w_o$ | 0.0619 | 0.0199 | 0.0778 | 0.1000 | 0.0862 | 0.3103 | 0.1283 | 0.0922 | 0.0340 | 0.0894 |
| $w_s$ | 0.0294 | 0.0235 | 0.0797 | 0.0490 | 0.1549 | 0.1549 | 0.1678 | 0.1549 | 0.1328 | 0.0531 |
| $w$ | 0.0456 | 0.0217 | 0.0787 | 0.0745 | 0.1205 | 0.2326 | 0.1481 | 0.1235 | 0.0834 | 0.0712 |

Since the final weights of the criteria have been calculated, the research's next step is to determine which EVs are most preferable for meeting the demands of the taxi service in the Brčko District. The phases of the MABAC method are executed in order to create the ranking order. The data is normalized once the initial decision-making matrix is formed. The normalized decision matrix will then be weighted (Expression no. 10). The border approximate area matrix's value (*G*) is then calculated. This value is calculated as the geometric mean, and the deviation from it is calculated. The geometric mean of this value is computed, and the deviation from it is calculated. The sum of these differences is used to calculate the final ranking order (Table 8). The best-ranked alternative, based on the MABAC method, is A11, followed by A6, and the worst-ranked alternative after applying the method is A2. In this respect, it was revealed that purchasing a Volkswagen ID.3 Pro (A11) or a Renault Mégane E-Tech EV40 (A6) is the best choice for taxi drivers in the Brčko District of Bosnia and Herzegovina.

**Table 8.** Alternative ranking order.

| Alternative | Qi | Rank |
|---|---|---|
| A1 | 0.0681 | 5 |
| A2 | −0.1671 | 11 |
| A3 | 0.0721 | 3 |
| A4 | −0.0495 | 10 |
| A5 | −0.0401 | 9 |
| A6 | 0.1353 | 2 |
| A7 | 0.0717 | 4 |
| A8 | −0.0348 | 8 |
| A9 | 0.0099 | 7 |
| A10 | 0.0643 | 6 |
| A11 | 0.2055 | 1 |

In order to examine the influence of the criteria on the final decision, a sensitivity analysis was then conducted. The results of the sensitivity analysis (Figure 1) show that alternative A11 is sensitive to the change of one criterion, namely criterion C5 (battery capacity), when only 25% of the original weight of that criterion is taken into account, compared to 10% of the weight of that criterion. It implies that one of the primary main contributors to the A11 alternative's ranking as the best in as many as 58 scenarios is its battery capacity. It is noteworthy to note that alternative A6 showed exceptional sensitivity to reducing the weight of criteria C6, or charge time, even if it typically came in at second place (52 scenarios) when implementing scenarios 34 to 36. One of the reasons why this alternative is typically placed in second place is charge time because it has a fast charging time on traditional outlets—much better than the other alternatives. It is also important to mention the A8 alternative, which demonstrated a dependence on the C7 criterion, the fast charge time criterion. When not having access to a fast charger, this alternative would rank considerably higher.

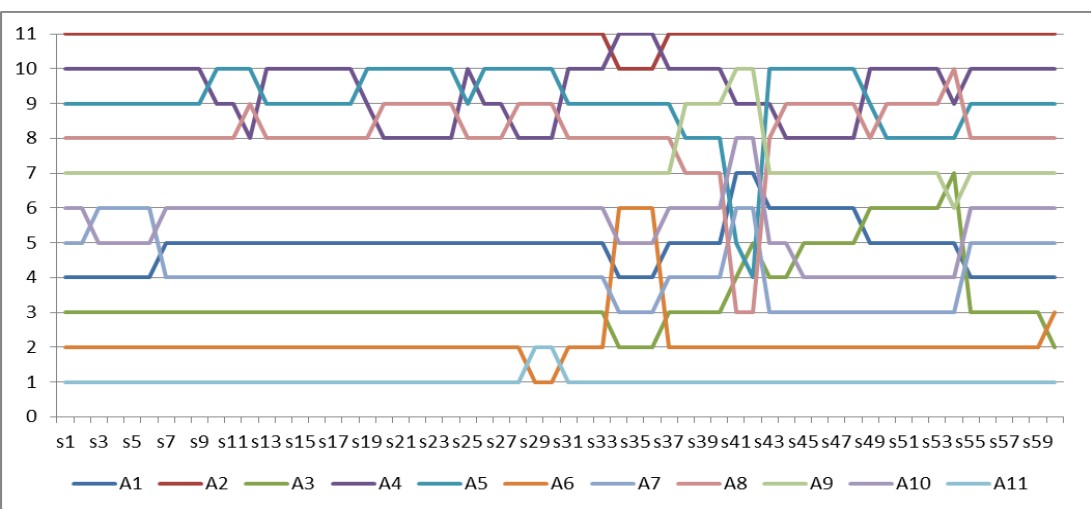

**Figure 1.** The sensitivity analysis's results.

## 5. Discussion

Using an EV has become the ideal alternative to traditional internal combustion engine vehicles. Environmental trends worldwide have been influenced by the fact that they are increasingly turning to alternate forms of transportation, namely, the adoption of electric motors. In various regions of the world, the adoption of electric vehicles in taxi services has become a commonplace event [18]. As a result, numerous governments are considering making EVs mandatory for taxi services. Some governments are going even farther, such as introducing autonomous taxis [19].

As a developing country, Bosnia and Herzegovina (BiH) is not currently considering mandating EVs for taxi services. It is important to understand that only 50 EVs were imported into Bosnia and Herzegovina in 2021. Thus, research on the taxi service in Bosnia and Herzegovina conducted in this paper is crucial in order to determine which EVs are currently the most appropriate for the taxi service.

Which EV would be the most ideal choice for taxi services was examined in this paper using the example of taxi drivers' demands in the Brčko District. According to the responses, the most important criteria for choosing an EV, in their opinion, are the fast charge time, followed by battery capacity, charge time, and range. Even though the research was set under different conditions, along with the fact that these criteria are also the most significant EV constraints [22], the selected criteria were in compliance with the reviewed literature [27–42]. Taxi drivers would prefer an EV that can be charged quickly and that can cover a considerable distance on a single charge. In this manner, they would optimize their business. They would experience fewer or no disruptions in their EV battery charging needs. However, to introduce EVs into taxi services, a good network of battery charging stations must be established [17]. Fast chargers are mostly unavailable to taxi drivers in Bosnia and Herzegovina's Brčko District since the network of battery charging stations is insufficiently established. In most cases, taxi drivers would be forced to charge their EVs at home using traditional outlets. To enable the use of EVs for taxi services in Bosnia and Herzegovina, investments in charging stations must first be made. This is also one of the most significant limits in the use of EVs for taxi service purposes [22]. In order to motivate taxi drivers to use EVs, it is, therefore, important to first establish a network of charging stations. Charging networks should be established not only for taxi drivers but also for other EV users; this would promote the usage of EVs among all individuals. The usage of EVs will be kept to a minimum in the absence of an appropriate infrastructure [44].

The results of criteria weighting using the MSDM method revealed that criterion C6 (charge time) has by far the highest weight when compared to other criteria. This is mainly attributable to the highest number of variances in the values of this criterion. The charging time with a traditional outlet ranges from 135 to 765 min. The charging time with fast

chargers, on the other hand, varies far less and spans from 26 to 47 min. The maximum speed that EVs can develop, on the other hand, shows the least variance and so received the least weight. With the exception of criterion C6 (charge time), the weights of the criteria calculated by the SWARA and the MSDM method do not differ significantly.

The ranking of the commercially available EVs was completed using the MABAC method [28,34], which reported that the best indications are of alternative A11, i.e., the Volkswagen ID.3 Pro. Similar outcomes were found in the research [27], where, in addition to the Volkswagen ID.3 Pro S, the Nissan LEAF e+ was also suggested. Since the other research findings included more expensive models, their rankings differed [46]. What distinguished this vehicle from others was not only its battery size but also its acceleration and maximum speed. Furthermore, the charging speeds with a conventional outlet and a fast charger are similar to those of other EVs. As a result, despite the fact that this vehicle is the most expensive on the market, it took first place in the ranking. The sensitivity analysis that was conducted supported these results.

Based on the research and results, it has been proven that all EV batteries and charging systems must be improved/established as these criteria are the things that distinguish EVS from one another in specific conditions in this case. Then, it will be essential to reach the maximum distance possible on a single battery charge. These are only a few directions for future research into the field of EV. Besides this, the prices of EVs are higher than the prices of traditional internal combustion vehicles. However, the market price of internal combustion vehicles is increasing, and fewer and fewer buyers are opting to purchase these vehicles.

## 6. Conclusions

The results of the research indicated that for the use of EVs in taxi services, it is crucial that these vehicles have the highest achievable range on a single charge and the fastest possible battery recharge rates. These are the most significant constraints of EVs. The ranking results of EV alternatives revealed that alternative A11 (Volkswagen ID.3 Pro) achieved the best results, while alternative A2 (Nissan Leaf) achieved the poorest. The sensitivity analysis was used to validate the results.

However, this research had certain limitations. The criteria used to rank the alternatives are the first constraint. The focus of this research was on the technical characteristics of EVs rather than the actual experiences with these vehicles, given that the EV market in Bosnia and Herzegovina has not yet developed to its full potential. In future research, taxi drivers with EV expertise must be included in the study, in order to provide the necessary data on the advantages and limitations of EVs in taxi services. The alternatives selected represent the research's limitations, in addition to the criteria. When deciding on alternatives with taxi drivers, numerous criteria were used, including price, the number of people that may be transported, the availability of authorized services, and expertise with those manufacturers. This research adopted 11 alternatives based on those criteria. Other alternatives, particularly novel EV concepts that are not currently on the market, must be considered in future research.

This research demonstrates how to select EVs for the purposes of taxi services. These services are highly significant in cities, particularly for local and tourist transportation. The need to substitute taxi service vehicles with ones that have the least negative effects on the environment emerges as a result, and, to motivate taxi drivers to adopt EVs as their service vehicle, a network of charging stations must be established, with a side note that government subsidies that encourage the use of EVs as light-duty vehicles might play a significant role.

**Author Contributions:** Conceptualization, A.Š. and A.P.; methodology, A.P.; software, A.P.; validation, A.Š. and D.B.; formal analysis, D.B.; investigation, A.P.; resources, A.P. and A.Đ.; data curation, A.P.; writing—original draft preparation, A.Š. and A.P.; writing—review and editing, A.Š.; visualization, A.Š. and A.P.; supervision, D.B.; project administration, A.Đ.; funding acquisition, A.Đ. All authors have read and agreed to the published version of the manuscript.

**Funding:** This research received no external funding.

**Institutional Review Board Statement:** Not applicable.

**Informed Consent Statement:** Not applicable.

**Data Availability Statement:** Not applicable.

**Conflicts of Interest:** The authors declare no conflict of interest.

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
