# Peer review of "Electric Vehicles Selection Based on Brčko District Taxi Service Demands, a Multi-Criteria Approach"

_urbansci, doi:10.3390/urbansci6040073_

Round 1
Reviewer 1 Report
This study seems interesting from the title. Undoubtedly, Electrical vehicles have a prominent role in the reduction of greenhouse gases. Fortunately, this paper is a good addition to the knowledge of electric vehicle studies. There is a need to consider some of the latest peer-reviewed research articles with respect to electrical vehicles specifically in the year 2022. Some suggestions to improve the manuscript are highlighted below. The overall manuscript is well articulated.
In the Abstract section, there are abbreviations/acronyms; SWARA, MSDM, MABAC and EV. Initially, there must be a complete form of a word or phrase and an abbreviation/acronym should be in the brackets.
- Page 2, Line 66, there is an abbreviation MCDM, please provide the full form of words here and then the abbreviation, as discussed above.
- Page 3, line 130, the spell of approach is wrong, correct it accordingly.
- Page 6, line 212, there is realculating, correct the spell.
Author Response
Dear Reviewer 1,
We would like to thank you for the opportunity to revise our manuscript 1952926. Thank you for your expertise, time, effort and utterly constructive critique on the manuscript.
After taking into consideration all the suggestions, we believe and hope that we have accomplished necessary modifications to the manuscript. We will try to give answers / replies to all of the individual tasks hereafter. Separate document, with highlighted changes (yellow) is provided, along with revised manuscript.
- Acknowledged. Reference list is updated
- Acknowledged. Abstract is updated with complete forms, and abbreviations are introduced in the paper itself.
- Acknowledged and corrected. (Former Page 2, Line 66)
- Acknowledged and corrected. (Former Page 3, line 130)
- Acknowledged and corrected. (Former Page 6, line 212)
We hope that the revision we have made will suffice and that the revised manuscript will be accepted for publishing in Urban Science. We highly appreciate your interest in our research and we would be happy to consider further improvements if necessary.
Kind regards,
Authors.

Author Response
Dear Reviewer 2,
We would like to thank you for the opportunity to revise our manuscript 1952926. Thank you for your expertise, time, effort and utterly constructive critique on the manuscript.
After taking into consideration all the suggestions, we believe and hope that we have accomplished necessary modifications to the manuscript. We will try to give answers / replies to all of the individual tasks hereafter. Separate document, with highlighted changes (green) is provided, along with revised manuscript.
Abstract:
Acknowledged. Abstract is updated with complete forms, and abbreviations are introduced in the paper itself.
LN14: Explained. The sentence is rewritten. Foundation was used as a term for backbone. We hope that change clarifies.
LN22: Acknowledged. “According to the research and led by battery capacity criterion and it’s values, the alternative Volkswagen ID.3 Pro has the best results…”
Introduction:
Acknowledged. The second national climate pledge to the United Nations Framework Convention on Climate Change (UNFCCC) was introduced in this section.
L52: Acknowledged. Thank you! This is unfortunately true, but a whole other issue on a global level.
L66: Acknowledged and corrected. Referring to L19 (MSDM) is not the case.
L71 – L81: Acknowledged and corrected.
Literature Review:
Section 2.1: Acknowledged and corrected. Thank you! This section did leave a sense of being fragmented. As per your suggestion we have reorganized (rewritten) it.
Section 2.2: Acknowledged and corrected. We hope that Table 1 footer will provide a bit of explanation and that our mark after will suffice to this section.
Methodology:
Acknowledged. A paragraph was added to the beginning of the Methodology section.
Acknowledged. References related to the phases of the research were added.
L164: Acknowledged. Further explanations added.
Table 2: C1. Acknowledged and corrected. True! Thank you! Corrections were made throughout the manuscript as a consequence, regarding the calculations and results.
L196: Acknowledged. Further explanations added.
Results:
Table 3 & 4: Acknowledged. Table 3 and Table 4 were updated per suggestion.
L310: Acknowledged and corrected.
L357: Acknowledged. Thank you! General framework of sensitivity analysis does belong to methodology, and thus it is moved to the Phase 7 of the Methodology section. We hope that you will agree with us that the Results section is now more focused on the actual results.
Discussion:
L404: Acknowledged and corrected.
Acknowledged and revised with additional paragraphs. .
L419: Acknowledged and paraphrased accordingly.
We hope that the revision we have made will suffice and that the revised manuscript will be accepted for publishing in Urban Science. We highly appreciate your interest in our research and we would be happy to consider further improvements if necessary.
Kind regards,
Authors.

Round 2
Reviewer 2 Report
Thank you very much for considering my comments and reviewing your paper accordingly. I can see that you have put a lot of effort into your revisions and I think, as a result, the paper has improved substantially. It is also much easier to read due to the abbreviations in the tables and the restructuring. I think that the paper would be ready to be published in the current format.
The only thing that could still be improved in my opinion (but I would leave that up to the editor) is the discussion. In my opinion, it still fails to contrast or discuss your findings with those of previous studies (especially regarding important selection criteria for EV purchases) to highlight the new contributions and innovations of your study.
Author Response
Dear Reviewer 2,
We would like to thank you for the opportunity to revise our manuscript 1952926, this time with minor changes that are required.
We believe that we have accomplished necessary modifications to the manuscript and the changes could be easily observed (green highlights and green underlines).
We hope that the changes made according to the request for the minor revision will suffice and that the revised manuscript will be accepted for publishing in Urban Science.
We thank you again for your interest in our research and highly appreciate your feedback in all.
Kind regards,
Authors.